# Counterfactual Outcome Estimation in Time Series via Sub-treatment Group Alignment and Random Temporal Masking

## Abstract

Estimating counterfactual outcomes in time series from observational data is important for effective decision-making in many fields, such as determining the optimal timing for a medical intervention. However, this task is challenging, primarily because of the unobservability of counterfactual outcomes and the complexity of confounding in time series. To this end, we introduce a representation learning-based framework for counterfactual estimation in time series with two novel techniques: **Sub-treatment Group Alignment** (SGA) and **Random Temporal Masking** (RTM). The first technique focuses on reducing confounding at each time point. While the common approach is to align the distributions of different treatment groups in the latent space, our proposed approach, SGA, first identifies *sub-treatment groups* through Gaussian Mixture Models (GMMs) and subsequently aligns the corresponding sub-groups. We demonstrate that, both theoretically and empirically, SGA achieves improved alignment, thus leading to more effective deconfounding. The second technique, RTM, masks covariates at random time steps with Gaussian noises. This approach promotes the time series models to select information not only important for the outcome estimation at current time point but also crucial for the time points in the future where the covariates are masked out, thus preserving the *causal information* and reducing the risk of overfitting to factual outcomes. We observe in experiments on synthetic and semi-synthetic datasets that applying SGA and RTM individually improves counterfactual outcome estimation, and when combined, they achieve state-of-the-art performance.

## 1 Introduction

Estimating causal effects in time series is important in various fields such as healthcare, politics, and economics (Morid et al., 2023; Freeman, 1983; Bisgaard & Kulahci, 2011). For example, consider the treatment of *Ductal Carcinoma In Situ* (DCIS) where the timing of surgical intervention is critical to the treatment effect: if surgery is too late, the cancer may progress to an invasive stage; if conducted too early, the procedure may be unnecessarily invasive (Grimm et al., 2022).

Motivated by this, we explore **counterfactual outcome estimation in time series from observational data**. The success of causal inference in time series relies on **effective reduction of time-dependent confounding**. However, this task is challenging, primarily because of the unobservability of counterfactual outcomes and the complexity of confounding in time series. A well-established group of approaches for reducing confounding in **static** causal inference is to minimize the upper bound of the counterfactual estimation error (Johansson et al., 2016; 2022; Li & Fu, 2017; Yao et al., 2018), which can be decomposed into two key components: (i) the *factual loss* and (ii) the *statistical discrepancy between treated and control groups* in the learned representation space. Algorithmically, these methods minimize the prediction error of the factual outcomes while **aligning the two treatment groups in the latent space**. By ensuring that the representations of two treatment groups are brought closer together, they provably reduce the bias introduced by confounders (Johansson et al., 2022). Building on this idea to reduce confounding for time series, existing approaches aim to learn representations that remain invariant to the treatment assignment **at each time step** (Bica et al., 2020; Melnychuk et al., 2022). However, in practice, with adversarial training, they typically result in optimizing relatively **loose upper bounds** on the counterfactual error at individual time

steps (Arjovsky & Bottou, 2017). Moreover, ***the error can accumulate over time steps*** and cause compromised estimation of long-term effects.

To this end, we provide two novel contributions that can be added to many current representation learning-based frameworks for counterfactual estimation on time series to tighten the upper bound and provide improved estimation: *Sub-treatment Group Alignment* (SGA) and *Random Temporal Masking* (RTM). Specifically, our techniques improve the existing approaches on two dimensions:

- SGA ***improves the alignment at each individual time point*** by first identifying *sub-treatment groups* and subsequently aligning the corresponding sub-groups.

- RTM ***blocks the accumulation of error*** by randomly selecting time points and masking the covariates at these time points with Gaussian noises.

**Sub-treatment Group Alignment (SGA).** SGA first identifies ***sub-treatment groups in the representation space*** through Gaussian Mixture Models (GMMs), and subsequently aligns the corresponding sub-groups of different treatment groups. See Figure 1 for a visual illustration. On an intuition level, alignment of sub-groups enables a ***more refined alignment*** of treatment groups, thus more effectively reducing confounding. In Section 4, we establish that sub-group alignments indeed lead to a tighter bound on the counterfactual estimation error. This allows us to ***reduce the estimation error more effectively than existing methods***.

**Random Temporal Masking (RTM).** While SGA addresses confounding at ***individual time points***, RTM enhances the model's ability to generalize ***across time series***. Inspired by masked language modeling, RTM uses random covariate masking, where ***the input covariates at randomly selected time points are replaced with Gaussian noise during training***. There are multiple perspectives to understand the benefits of RTM:

- At the time points where the input covariates are replaced with noise, the time series models are forced to extract useful information from previous time steps to predict the factual outcome in the future. In other words, we encourage the model to ***focus on the causal relationships that span across time***, leading to better counterfactual predictions.

- RTM can prevent model from becoming overly reliant on the information from the current time points, thus ***reducing overfitting to the factual distribution***.

- RTM resets the time series by completely replacing the covariates at selected time steps with noise, ***blocking the accumulation of error***.

**Empirical Validation.** Our approach is general and can be built upon and adapted to the objective function of a wide range of time series estimation methods, offering ***broad applicability***. We validate this through comprehensive experiments on synthetic and semi-synthetic datasets, demonstrating state-of-the-art performance in counterfactual outcome estimation.

**Organization.** We first formally define the problem in Section 2 and review related works in Section 3. Then in Section 4, we theoretically establish how sub-treatment group alignment achieves improved alignment, thus motivating our SGA technique. In Section 5, we present our framework with SGA and RTM as components. Experimental results in Section 6 show that applying SGA and RTM individually enhances performance, and when combined, they achieve state-of-the-art results.

## 2 PROBLEM SETUP

**Notations.** We use upper-case letters (e.g., $A, Y$) for scalar random variables and lower-case letters (e.g., $a, y$) for their corresponding realizations. Multi-dimensional random variables and realizations are denoted using bold fonts (e.g., $\boldsymbol{X}$ and $\boldsymbol{x}$).

**Observational Data.** Following the setup in Melnychuk et al. (2022); Bica et al. (2020); Li et al. (2020), we consider a dataset containing $N$ samples. Observations are recorded over $T$ time steps, i.e., $t = 1, ..., T$. At each time step $t$, a discrete treatment $A_t \in \mathcal{A} = \{a_0, a_1, ..., a_{|\mathcal{A}|-1}\}$ is assigned to the sample. Thus, for each sample $i$, we observe time-varying covariates $\mathbf{X}_t^{(i)} \in \mathbb{R}^d$, the factual treatment $A_t^{(i)}$, and the outcome $Y_t^{(i)}$ associated with the factual treatment.

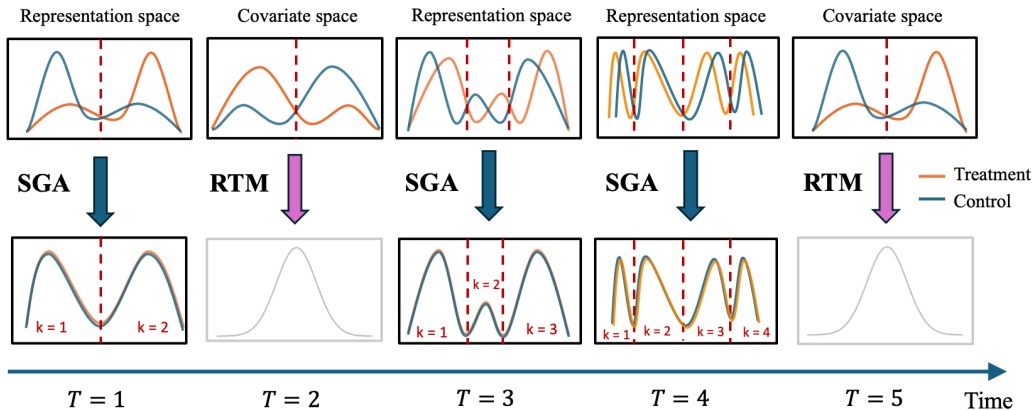

Figure 1: **Conceptual Overview of Our Method.** This figure illustrates our approaches Sub-treatment Group Alignment (SGA) and Random Temporal Masking (RTM) to improve counterfactual outcome estimation in time series from observational data. Here, k represents the sub-treatment group index. For simplicity, only two treatment groups are shown: treatment and control.

We use the following notation to represent the process up to time step $t$ for each unit $i$:

$$\bar{\mathbf{H}}_t^{(i)} = \left\{ \bar{\mathbf{X}}_t^{(i)}, \bar{\mathbf{Y}}_t^{(i)}, \bar{\mathbf{A}}_{t-1}^{(i)}, \mathbf{V}^{(i)} \right\}, \text{ where:}$$

- $\bar{\mathbf{X}}_t^{(i)} = \{\mathbf{X}_s^{(i)} : s \leq t\}$ denotes the sequence of time-varying covariates up to time $t$,

- $\bar{\mathbf{Y}}_t^{(i)} = \{Y_s^{(i)} : s \leq t\}$ represents the sequence of observed outcomes up to time $t$,

- $\bar{\mathbf{A}}_{t-1}^{(i)} = \{A_s^{(i)} : s \leq t - 1\}$ is the sequence of treatments up to time $t - 1$,

- $\mathbf{V}^{(i)} \in \mathbb{R}^p$ denotes the static covariates (those that do not change over time).

**Objective.** Given the process up to current time $t$ and assuming a specific treatment sequence $\mathbf{a}_{t:t+\tau-1}^{(i)}$ from time $t$ to $t + \tau - 1$ applied to sample $i$, ***our goal is to estimate***, for each unit $i$, ***the future outcome at time step*** $t + \tau$. That is, $\tau$ time steps after the current time $t$. To ensure that these counterfactual outcomes are identifiable, we follow the *potential outcomes framework* and make several standard assumptions to support identifiability (Rosenbaum & Rubin, 1983; Rubin, 2005). Due to space constraint, details on the assumptions are provided in Appendix A.

Specifically, we aim to estimate:

$$\mathbb{E}\left[ Y_{t+\tau}^{(i)}\left(\mathbf{a}_{t:t+\tau-1}^{(i)}\right) \,\middle|\, \bar{\mathbf{H}}_t^{(i)} \right], \tag{1}$$

where $Y_{t+\tau}^{(i)}\left(\mathbf{a}_{t:t+\tau-1}^{(i)}\right)$ denotes the potential outcome at time $t + \tau$ for unit $i$ under the treatment sequence $\mathbf{a}_{t:t+\tau-1}^{(i)}$.

## 3 RELATED WORK

We review the most relevant work below and provide a comprehensive discussion in Appendix C.

**Estimating counterfactual outcomes under static scenarios.** Many methods have been proposed to learn a *balanced* representation that aligns the distributions across treatment groups, effectively addressing confounding in static settings. A foundational work in this area, CFRNet proposed by Shalit et al. (2017), establishes a counterfactual error bound illustrating that the expected error in estimating individual treatment effects (ITE) is bounded by the sum of its standard generalization error and the discrepancy between treatment group distributions induced by the representation. This concept has been further explored in several subsequent studies on deep causal inference (Yao et al.,

2018; Kallus, 2020; Du et al., 2021). However, these methods primarily focus on static data, and their approach of aligning overall treated and control group distributions may not sufficiently adaptable to time-series data (Hernán et al., 2000; Mansournia et al., 2012), where time-dependent confounders make it difficult to disentangle the true effect of a treatment from these caused by the confounding variables.

**Estimating counterfactual outcomes over time.** Estimating counterfactual outcomes in time-series data is challenging due to time-varying confounders. Traditional methods such as G-computation and marginal structural models (Robins, 1986; Robins et al., 2000; Hernán et al., 2001; Robins & Hernan, 2008; Xu et al., 2016) often lack flexibility for complex datasets and rely on strong assumptions. To address these limitations, researchers have developed models that build on the potential outcomes framework initially proposed by Rubin (1978) and extended to time series by Robins & Hernan (2008). Notable among recent methods are Recurrent Marginal Structural Networks (RMSNs) (Lim, 2018), G-Net (Li et al., 2020), Counterfactual Recurrent Networks (CRN) (Bica et al., 2020), and the Causal Transformer (CT) (Melnychuk et al., 2022), which use approaches such as propensity networks and adversarial learning to mitigate the effects of time-varying confounding. However, practical challenges with adversarial training can affect the stability of causal effect estimations. Specifically, training adversarial networks can be challenging due to issues such as mode collapse and oscillations (Liang et al., 2018). Additionally, adversarial training minimizes the Jensen-Shannon divergence (JSD) only when the discriminator is optimal (Arjovsky & Bottou, 2017), which may not always be achievable in practice; even when the discrminator is optimal, using JSD optimizing relatively loose upper bounds on the counterfactual error. To address these challenges, we propose using the Wasserstein-1 distance and provides stronger theoretical guarantees (Redko et al., 2017; Mansour et al., 2012).

**Masked language modeling.** Masked language modeling (MLM) is a common self-supervised pre-training technique for large language models. It operates by randomly masking certain words or tokens in the input, with the model trained to predict the masked tokens. BERT (Devlin, 2018) is the most well-known model that uses this technique. Recent studies have also demonstrated the effectiveness of MLM in enhancing generalization across sequence-based tasks. For example, Chaudhary et al. (2020) shows that when combined with cross-lingual dictionaries, MLM improves predictions for the original masked word and also generalizes to its cross-lingual synonyms. Inspired by the success of masking strategies in language models, we introduce Random Temporal Masking (RTM) for time-series data. Unlike MLM, which focuses on predicting the masked inputs, RTM encourages the model to focus on information that is crucial for both the current time point and future time points, preserve causal information, and reduce the risk of overfitting to factual outcomes.

## 4 THEORETICAL MOTIVATION FOR SUB-TREATMENT GROUP ALIGNMENT

This section provides a theoretical motivation for our proposed Sub-treatment Group Alignment (SGA) method, rigorously illustrating that *aligning sub-treatment groups in the latent space leads to more effective deconfounding in estimating counterfactual outcomes* over time series.

**From Static to Time Series.** We first note that *SGA is in essence an improved alignment method for static causal inference* problems where the total number of time steps is 1. In this setting, alignment of treatment groups has proven effective in reducing confounding. By aligning the corresponding sub-treatment groups, *SGA results in more refined alignment* and thus more effective confounding reduction. Building on the idea of alignment in static setting, *existing approaches for time series align the covariates at individual time steps*. In other words, these approaches consider the confounding problems at individual time steps to be static problems, and align them individually. To this end, replacing existing alignment method at each time step with SGA *improves alignment at every time step*, *leading to more effective confounding reduction for the whole time series*.

**Section Organization.** Given that existing approaches for time series consider alignments at varying time steps as individual static problems and we aim to establish that SGA improves alignment at every time step, it is *sufficient to consider static settings*. Thus, in Section 4.1 we briefly review *representation learning-based models* which are based on the idea of alignment and *why alignment helps preventing bias from confounders* in the static setting. In Section 4.2, we *theoretically establish that SGA indeed improves alignment* in the static setting, implying that it improves alignment

for existing approaches for time series at each individual time step. It follows naturally that SGA leads to overall improvement for the time series.

## 4.1 Alignment for Static Setting

Since there is only one time step $t = 1$ in the static setting, we will omit all notations about the time step for clarity. We will use the Wasserstein-1 distance $W_1$ to measure the statistical discrepancy between two random variables. Due to space constraint, we defer mathematical definition of $W_1$ to Appendix D.11.

**Representation Learning-based Models.** Let $\Phi : \mathcal{X} \to \mathcal{R}$ be a representation-learning function and $h : \mathcal{R} \times \{0, 1\} \to Y$ be an hypothesis. We have $h(\Phi(x), a)$ as a predictor for an individual $x$'s potential outcome under treatment assignment $a$. The goal is to find a pair of $(h, \Phi)$ that optimizes both the *factual loss* $\epsilon_F^\star(h, \Phi)$ and *counterfactual loss* $\epsilon_{CF}(h, \Phi)$, which are defined in Appendix D.2 and D.9 due to space constraint. Note that low factual and counterfactual losses are both necessary and sufficient conditions for accurate potential outcome prediction (Aloui et al., 2023).

**Counterfactual Error Estimation.** However, the ***counterfactual loss*** $\epsilon_{CF}(h, \Phi)$ ***cannot be directly optimized*** because the counterfactual outcomes are not observed in real-world scenarios. To this end, a group of well-established approaches ***minimize upper bounds of*** $\epsilon_{CF}(h, \Phi)$. These approaches are mainly based on the following result from Shalit et al. (2017), which provides an upper bound for $\epsilon_{CF}(h, \Phi)$ with observable quantities.

**Theorem 4.1** (Simplified Lemma A8 from Shalit et al. (2017), complete version provided in Appendix D.10.). *Let $\Phi : \mathcal{X} \to \mathcal{R}$ be a one-to-one and Jacobian-normalized representation function. Let $h : R \times \{0, 1\} \to Y$ be a hypothesis with Lipschitz constant:*

$$\epsilon_{CF}(h, \Phi) \leq \epsilon_F^\star(h, \Phi) + 2 \cdot B_\Phi \cdot W_1(p_\Phi^0, p_\Phi^1), \qquad (2)$$

*where $B_\Phi$ is a constant and $p_\Phi^a$ is the distribution of the random variable $\Phi(X)$ conditioned on $A = a$, that is, representations for individuals receiving treatment $a \in \{0, 1\}$.*

**Motivation for Alignment.** This theorem implies that a model $(\Phi, h)$ has low counterfactual error if *(i)* it has ***low factual error*** (which can be easily achieved by minimizing the prediction error on the observational data) and *(ii)* the covariates of individuals from distinct treatment groups are ***statistically similar to each other in the latent (representation) space***. Motivated by these, representation learning-based methods aim to align the treated and control groups in the latent space while minimizing the factual error. In particular, successful alignment and low factual error guarantee a small value for the upper bound in Equation (2), implying the model has low counterfactual error. However, in practice, ***the error bound may be loose***, leaving the model performance suboptimal.

## 4.2 Benefits of Sub-treatment Group Alignment

To this end, we propose to ***use the sub-treatment group structures to achieve tighter counterfactual error bound***, thus supporting more effective alignment.

**Sub-treatment Groups.** We assume that *each treatment group is a mixture of $K$ sub-treatment groups in the latent space*, and that *the sub-treatment groups across different treatment groups correspond to one another*. For example, in medical studies, patients ***may naturally form sub-groups before the beginning of experiments based on latent variables such as demographic characteristics or genetic factors***. Consider a scenario where patients are sub-grouped according to age (e.g., children, adults, seniors), gender, or genetic markers that influence their response to treatment. Even though these patients receive different treatments, the underlying characteristics defining the sub-groups are consistent across treatment groups. By aligning these corresponding sub-groups in the latent space, we can more effectively account for hidden confounders like genetic predispositions or socio-demographic factors, leading to more accurate estimation of treatment effects.

Specifically, we have:

$$p_\Phi^0 = \sum_{k=1}^K w_k^0 P_{\Phi,k}^0, \quad p_\Phi^1 = \sum_{k=1}^K w_k^1 P_{\Phi,k}^1,$$

where for $a \in \{0, 1\}$, $w_k^a$ represents the proportion of the $k$-th sub-group in treatment group $a$, and $P_{\Phi,k}^a$ denotes the distribution of the representations of the individuals in the $k$-th sub-group under treatment $a$.

**Sub-treatment Group Alignment (SGA).** SGA has the following alignment objective:

$$\sum_{k=1}^{K} w_k^1 W_1 \left( P_{\Phi,k}^0, P_{\Phi,k}^1 \right). \tag{3}$$

In particular, SGA minimizes the ***weighted sum*** of the Wasserstein distances between these ***corresponding sub-treatment groups***. By aligning on a sub-treatment group level, SGA achieves more refined alignment. Indeed, motivated by the generalization bound in the field of *domain adaptation* (Liu et al., 2023), we next prove in Theorem 4.2 that ***SGA is at least as tight as the original alignment*** under reasonable assumptions, thus resulting in more effective deconfounding.

**Theorem 4.2** (SGA Improves Generalization Bounds). *Under the following assumptions:*

***A1.*** *For all $k$, the sub-distributions $P_{\Phi,k}^0$ and $P_{\Phi,k}^1$ are Gaussian distributions with means $m_k^0$ and $m_k^1$, and covariances $\Sigma_k^0$ and $\Sigma_k^1$, respectively. The distance between corresponding sub-distributions is less than or equal to the distance between non-corresponding sub-distributions, i.e., $W_1(P_{\Phi,k}^0, P_{\Phi,k}^1) \leq W_1(P_{\Phi,k}^0, P_{\Phi,k'}^1)$ for $k \neq k'$.*

***A2.*** *There exists a small constant $\epsilon > 0$, such that $\max_{1 \leq k \leq K} (tr(\Sigma_k^0)) \leq \epsilon$ and $\max_{1 \leq k \leq K} (tr(\Sigma_k^1)) \leq \epsilon$.*

*Then the following inequalities hold:*

$$\epsilon_{CF}(h, \Phi) \leq \epsilon_F(h, \Phi) + 2B_\Phi \left( \sum_{k=1}^{K} w_k^1 W_1(P_{\Phi,k}^0, P_{\Phi,k}^1) \right) \quad and$$

$$\sum_{k=1}^{K} w_k^1 W_1(P_{\Phi,k}^0, P_{\Phi,k}^1) \leq W_1(p_\Phi^0, p_\Phi^1) + \delta_c,$$

*where $B_\Phi$ is the same constant in Theorem 4.1 and $\delta_c$ is $4\sqrt{\epsilon}$.*

*Proof of Theorem 4.2.* See in Appendix D.16. □

*Remark* 4.3. Theorem 4.2 proves that sub-treatment group alignment ***improves the original counterfactual error bound in Theorem 4.1*** by ***optimizing an upper bound that is at least as tight as the original bound***. In Appendix F.1.3, we provide empirical evidence that SGA indeed results in a much tighter upper bound compared to the original counterfactual error bound.

## 5 FRAMEWORK

We propose a framework for counterfactual estimation in time-series data that incorporates our two novel techniques: Sub-treatment Group Alignment (SGA) and Random Temporal Masking (RTM).

**Model Architecture.** Importantly, our framework is not restricted to any specific architecture and can be integrated with various representation-based approaches for causal inference in time series. Figure 2 illustrates our framework. In particular, we consider approches consisting of an ***time series encoder*** $\phi_E$, parameterized by $\theta_E$, which learns representations of the input time series data, and a ***regressor*** $f_Y$, parameterized by $\theta_Y$, which predicts the outcome at the next time point. We note that the encoder $\phi_E$ can be instantiated with any sequence model architecture, such as RNNs, LSTMs (Hochreiter, 1997), or transformers (Vaswani, 2017). In Section 6, we experiment with two such approaches *Causal Transformer* (Melnychuk et al., 2022) and *Counterfactual Recurrent Networks* (CRN) (Bica et al., 2020), which are well-established for causal inference in time series.

**Random Temporal Masking (RTM).** ***RTM is applied to the observational data before the training of models***. To implement RTM, we mask covariates at a set of randomly selected time steps by replacing them with Gaussian noise. The model is subsequently trained to predict the outcomes despite these masked covariates, encouraging it to focus on causal information that is robust over time. RTM also reduces the risk of overfitting to factual outcomes at those selected time steps because, after masking, the covariates at current time is independent of the outcome. This is ***particularly helpful when the potential outcomes at the current time steps are strongly correlated with current covariates*** because under these scenarios the models are inclined to heavily rely on current covariates, thus overfitting to the factual distribution.

**Objective Function.** Our framework optimizes the following objective function at each time step $t$:

$$\min_{\theta_Y, \theta_E} L_Y^t(\theta_Y, \theta_E) + \lambda L_D^t(\theta_E),$$

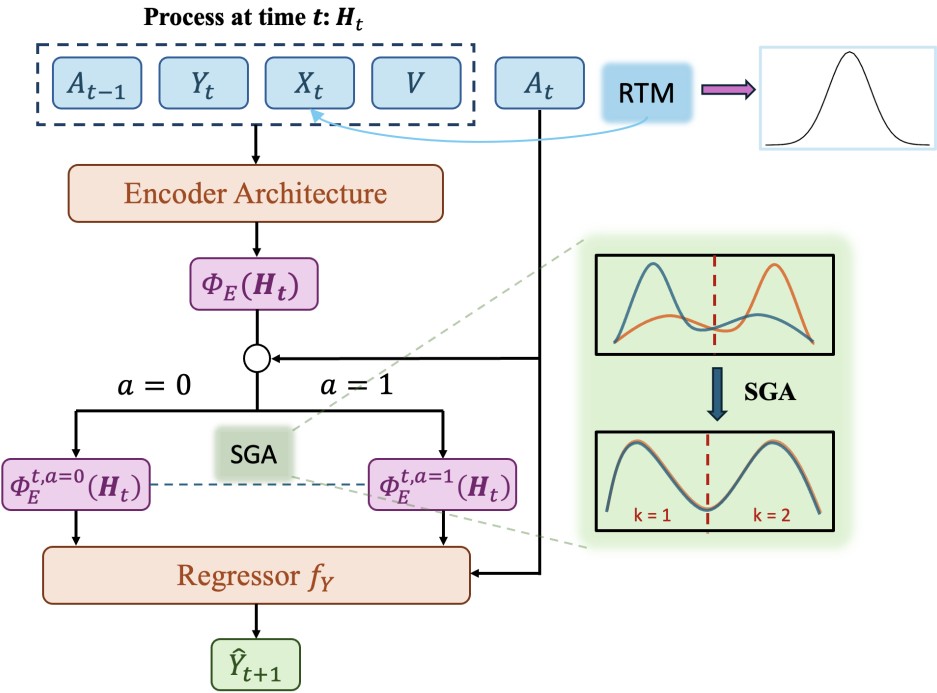

Figure 2: Overview of our method at each timepoint t - for simplicity, we only show binary treatment scenario. Our method is flexible, and can be integrated with many representation-based approaches for time-series causal inference, including CRN Bica et al. (2020) and CT Melnychuk et al. (2022), among others.

where $L_Y^t$ represents the ***factual outcome loss*** and $L_D^t$ denotes the ***SGA loss*** calculated with the Wasserstein-1 distance, balanced by $\lambda$. We next elaborate on them in detail.

**Factual Outcome Loss.** At each time step $t$, the model ***learns to predict the observed outcomes***, conditioned on $\mathbf{H}_t^{(i)}$ which contains the information from previous steps and the current covariates, by optimizing the following objective loss:

$$L_Y^t(\theta_Y, \theta_E) = \frac{1}{N} \sum_{i=1}^{N} (\ell(y_i^{t+1}, \hat{y}_i^{t+1})),$$

where $\hat{y}_i^{t+1} = f_Y\left(\phi_E\left(\mathbf{H}_t^{(i)}, A_t^{(i)}\right)\right)$, and $\ell$ denotes the loss function (e.g., mean squared error).

**SGA Loss.** Motivated by Section 4, our framework aligns the sub-treatment groups across distinct treatment groups. To this end, at each time step $t$ and for each treatment group $a$, we use Gaussian Mixture Models (GMMs) to cluster the individuals' features in the representation space into a pre-specifed $K$ sub-treatment groups. Let the random variable $\phi_E^{t,a,k}(\mathbf{H}_t)$ denote ***the representations of samples in the $k$-th sub-group of treatment group $a$ at time step $t$***.

To accommodate the applications with ***multiple treatment groups*** (more than two), we propose, for each time step $t$ and each corresponding sub-treatment group, to ***align the sub-treatment groups with the uniform mixtures of them***. That is, for all the $k$-th sub-treatment groups in all $|\mathcal{A}|$ treatment groups where $|\mathcal{A}|$ is the total number of treatments, we first create a mixture of them with uniform weights and align all of them with the uniform mixture. Note that by triangle inequality of the Wasserstein distance, this is a ***sufficient*** condition to align multiple groups well. Specifically, the SGA loss is defined as:

$$L_D^t(\theta_E) = \sum_{k=1}^{K} \sum_{a \in \mathcal{A}} w_k^{t,a} W_1(\phi_E^{t,a,k}(\mathbf{H}_t), \phi_E^{t,k}(\mathbf{H}_t)),$$

where $w_k^{t,a}$ represents the proportion of samples in sub-group $k$ of treatment group $a$, and $\phi_E^{t,k}(\mathbf{H}_t)$ is the uniform mixture of $\{\phi_E^{t,a,k}(\mathbf{H}_t)\}_{a\in\mathcal{A}}$. Note that all the quantities in $L_D^t(\theta_E)$ can be estimated from the observational data. We provide implementation details and our algorithm in Appendix E.

## 6 EXPERIMENTS

We conduct experiments on a ***fully-synthetic dataset*** and a ***semi-synthetic dataset*** to evaluate the effectiveness of our proposed methods.. The detailed experimental setup is provided in Appendix F. In our experiments, we demonstrate the effectiveness and flexibility of our proposed methods, SGA and RTM, by ***integrating them with existing state-of-the-art models***. Specifically, we incorporate our techniques into the architectures of the LSTM-based *Counterfactual Recurrent Networks (CRN)* (Bica et al., 2020) and the transformer-based *Causal Transformer (CT)* (Melnychuk et al., 2022). We observe that integration of SGA and RTM into CRN and CT improves their performance, establishing ***new state-of-the-art (SOTA) performance***.

**Baseline Methods**   We compare our methods against baseline approaches that have shown SOTA performance in the literature for time-series counterfactual outcome estimation. These include: Marginal Structural Models (Robins et al., 2000; Hernán et al., 2001), Recurrent Marginal Structural Networks (RMSNs) (Lim, 2018), G-Net (Li et al., 2020), Counterfactual Recurrent Networks (CRN) (Bica et al., 2020), and Causal Transformer (CT) (Melnychuk et al., 2022).

### 6.1 EXPERIMENTS WITH FULLY-SYNTHETIC DATA

We first consider a fully-synthetic benchmark frequently used in the counterfactual outcome estimation literature (Bica et al., 2020; Melnychuk et al., 2022). This dataset is generated with a *Pharmacokinetic-Pharmacodynamic* (PK-PD) model of tumor growth (Geng et al., 2017), allowing us to simulate treatment-response dynamics and varying levels of time-dependent confounding.

**Metric and Tasks**. Following Melnychuk et al. (2022), we assess the performance of our methods by computing the normalized Root Mean Squared Error (RMSE) between the true counterfactual outcomes and the estimated counterfactual outcomes on both ***one-step-ahead prediction*** and $\tau$-***step-ahead prediction tasks***. These evaluations are conducted under ***varying levels of time-varying confounding***, indexed by $\gamma$. Detailed information on dataset generation and hyperparameter settings is provided in Appendix F.1.

**Results Overview.** We first show that ***combining SGA and RTM achieves SOTA performance***, outperforming existing methods. We then demonstrate that ***applying SGA and RTM individually*** also improves counterfactual outcome estimation compared to baseline models.

### 6.1.1 COMBINED PERFORMANCE OF SGA AND RTM

As shown in Figure 3, applying SGA and RTM on top of CT and CRN ***significantly improves their performance compared to the vanilla models***, on *both* one-step-ahead and $\tau$-step-ahead prediction tasks. Furthermore, our methods also achieve *superior performance* compared to all other benchmark methods in almost all of the settings. Notably, ***our methods perform exceptionally well in scenarios with high levels of confounding***, indicating their ***effectiveness in deconfounding***. The performance of the benchmark methods is sourced from Melnychuk et al. (2022).

### 6.1.2 INDIVIDUAL PERFORMANCE OF SGA

We next evaluate the individual performance of SGA. As shown in Table 1, incorporating SGA into both CRN and CT achieves ***superior results compared to the vanilla models***. The introduction of SGA consistently improves prediction accuracy, with ***more pronounced improvements in settings with higher levels of confounding***. This supports our claim that SGA results in more refined alignment and thus more effective confounding reduction.

It is important to note that in scenarios with *no confounding ($\gamma = 0$)*, our methods do not perform as strongly. This is because aligning distributions across different treatment groups is unnecessary

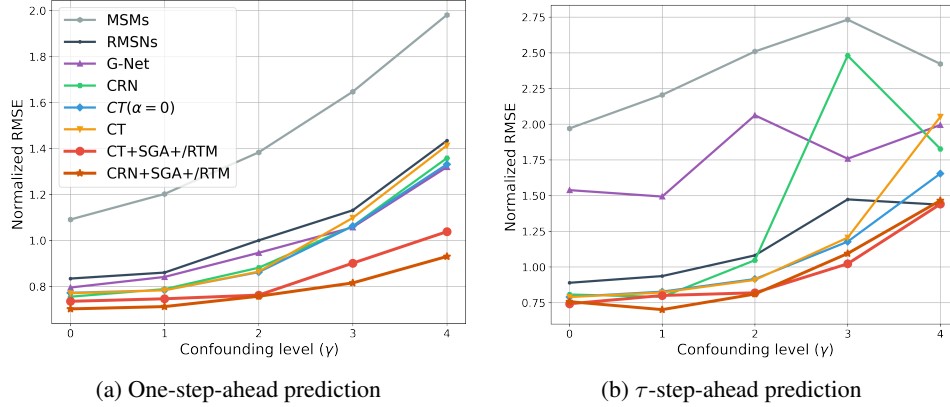

(a) One-step-ahead prediction          (b) $\tau$-step-ahead prediction

Figure 3: Performance comparison on (a) one-step-ahead prediction and (b) $\tau$-step-ahead prediction tasks under varying levels of time-varying confounding (indexed by $\gamma$). Our methods (CT+SGA+RTM and CRN+SGA+RTM) significantly outperform baseline models, especially in high-confounding scenarios. Note that CT ($\alpha = 0$) refers to CT without domain confusion loss to balance the representation.

Table 1: Normalized RMSE for one-step-ahead and $\tau$-step-ahead predictions on fully synthetic data, comparing vanilla CRN/CT with CRN/CT enhanced with SGA.

| $\tau$ | Method | $\gamma=0$ | $\gamma=1$ | $\gamma=2$ | $\gamma=3$ | $\gamma=4$ |
|---|---|---|---|---|---|---|
| | CRN | **0.755** | 0.788 | 0.881 | 1.062 | 1.358 |
| | CRN + SGA | 0.808 | **0.764** | **0.819** | **0.986** | **1.208** |
| $\tau=1$ | CT | **0.770** | 0.783 | 0.864 | 1.098 | 1.413 |
| | CT + SGA | 0.816 | **0.754** | **0.843** | **1.010** | **1.191** |
| | CRN | 0.671 | 0.666 | 0.741 | 1.668 | 1.151 |
| | CRN + SGA | **0.633** | **0.632** | **0.656** | **0.722** | **1.036** |
| $\tau=2$ | CT | 0.681 | 0.677 | **0.713** | 0.908 | 1.274 |
| | CT + SGA | **0.645** | **0.645** | 0.718 | **0.848** | **1.116** |
| | CRN | 0.700 | 0.692 | 0.818 | 1.959 | 1.360 |
| | CRN + SGA | **0.656** | **0.650** | **0.698** | **0.864** | **1.116** |
| $\tau=3$ | CT | 0.703 | 0.712 | 0.770 | 1.010 | 1.536 |
| | CT + SGA | **0.662** | **0.691** | **0.762** | **0.925** | **1.300** |
| | CRN | 0.734 | 0.722 | 0.898 | 2.201 | 1.573 |
| | CRN + SGA | **0.689** | **0.668** | **0.743** | **0.998** | **1.223** |
| $\tau=4$ | CT | 0.726 | 0.748 | 0.822 | 1.089 | 1.762 |
| | CT + SGA | **0.682** | **0.723** | **0.813** | **0.979** | **1.390** |
| | CRN | 0.769 | 0.755 | 0.976 | 2.361 | 1.730 |
| | CRN + SGA | **0.726** | **0.686** | **0.782** | **1.114** | **1.341** |
| $\tau=5$ | CT | 0.756 | 0.786 | 0.870 | 1.154 | 1.922 |
| | CT + SGA | **0.708** | **0.762** | **0.854** | **1.022** | **1.454** |
| | CRN | 0.807 | 0.790 | 1.047 | 2.480 | 1.827 |
| | CRN + SGA | **0.757** | **0.701** | **0.810** | **1.218** | **1.465** |
| $\tau=6$ | CT | 0.789 | 0.821 | 0.909 | 1.205 | 2.052 |
| | CT + SGA | **0.742** | **0.800** | **0.876** | **1.040** | **1.440** |

Table 2: Normalized RMSE for one-step-ahead and $\tau$-step-ahead predictions on fully synthetic data, comparing vanilla CRN/CT with CRN/CT enhanced with RTM.

| $\tau$ | Method | $\gamma=0$ | $\gamma=1$ | $\gamma=2$ | $\gamma=3$ | $\gamma=4$ |
|---|---|---|---|---|---|---|
| | CRN | 0.755 | 0.788 | 0.881 | 1.062 | 1.358 |
| | CRN + RTM | **0.702** | **0.712** | **0.757** | **0.815** | **0.930** |
| $\tau=1$ | CT | 0.770 | 0.783 | 0.864 | 1.098 | 1.413 |
| | CT + RTM | **0.735** | **0.746** | **0.762** | **0.901** | **1.038** |
| | CRN | **0.671** | **0.666** | 0.741 | 1.668 | **1.151** |
| | CRN + RTM | 0.705 | 0.674 | 0.745 | **0.990** | 1.153 |
| $\tau=2$ | CT | **0.681** | **0.677** | 0.713 | 0.908 | 1.274 |
| | CT + RTM | 0.686 | 0.677 | **0.693** | **0.785** | **1.004** |
| | CRN | **0.700** | 0.692 | 0.818 | 1.959 | 1.360 |
| | CRN + RTM | 0.726 | **0.687** | **0.791** | **0.893** | **1.219** |
| $\tau=3$ | CT | 0.703 | 0.712 | 0.770 | 1.010 | 1.536 |
| | CT + RTM | **0.691** | **0.697** | **0.720** | **0.856** | **1.194** |
| | CRN | **0.734** | **0.722** | 0.898 | 2.201 | 1.573 |
| | CRN + RTM | 0.756 | 0.724 | **0.862** | **0.973** | **1.377** |
| $\tau=4$ | CT | 0.726 | 0.748 | 0.822 | 1.089 | 1.762 |
| | CT + RTM | **0.707** | **0.735** | **0.752** | **0.921** | **1.362** |
| | CRN | **0.769** | **0.755** | 0.976 | 2.361 | 1.730 |
| | CRN + RTM | 0.783 | 0.765 | **0.907** | **1.041** | **1.474** |
| $\tau=5$ | CT | 0.756 | 0.786 | 0.870 | 1.154 | 1.922 |
| | CT + RTM | **0.725** | **0.765** | **0.787** | **0.968** | **1.522** |
| | CRN | 0.807 | **0.790** | 1.047 | 2.480 | 1.827 |
| | CRN + RTM | **0.802** | 0.796 | **0.934** | **1.094** | **1.541** |
| $\tau=6$ | CT | 0.789 | 0.821 | 0.909 | 1.205 | 2.052 |
| | CT + RTM | **0.745** | **0.800** | **0.819** | **1.022** | **1.663** |

**Note:** The values in **blue** indicate lower RMSE for CRN-based models, and values in **violet** indicate lower RMSE for CT-based models. The results demonstrate that both SGA and RTM consistently improve performance, especially in settings with higher levels of confounding (indexed by $\gamma$).

when there is no confounding. Consequently, introducing an extra alignment loss in such cases can interfere training, leading to suboptimal performance.

### 6.1.3 INDIVIDUAL PERFORMANCE OF RTM

We evaluate the individual performance of RTM by comparing the vanilla CRN and CT models against their counterparts enhanced with RTM. As shown in Table 2, introducing RTM consistently improves prediction accuracy, with ***more improvements in later time steps*** and settings with ***higher levels of confounding***. This confirms that RTM encourages the model to focus on causal relationships that span across time, and mitigating error accumulation.

Table 3: RMSE for one-step-ahead and $\tau$-step-ahead predictions on semi-synthetic data based on real-world medical data (MIMIC-III).

|  | $\tau=1$ | $\tau=2$ | $\tau=3$ | $\tau=4$ | $\tau=5$ | $\tau=6$ | $\tau=7$ | $\tau=8$ | $\tau=9$ | $\tau=10$ |
|---|---|---|---|---|---|---|---|---|---|---|
| MSMs | 0.37 | 0.57 | 0.74 | 0.88 | 1.14 | 1.95 | 3.44 | > 10.0 | > 10.0 | > 10.0 |
| RMSNs | 0.24 | 0.47 | 0.60 | 0.70 | 0.78 | 0.84 | 0.89 | 0.94 | 0.97 | 1.00 |
| G-Net | 0.34 | 0.67 | 0.83 | 0.94 | 1.03 | 1.10 | 1.16 | 1.21 | 1.25 | 1.29 |
| CRN | 0.30 | 0.48 | 0.59 | 0.65 | 0.68 | 0.71 | 0.72 | 0.74 | 0.76 | 0.78 |
| CRN + SGA + RTM | **0.27** | **0.43** | **0.52** | **0.58** | **0.62** | **0.65** | **0.67** | **0.69** | **0.72** | **0.73** |
| CT ($\alpha=0$) | **0.20** | **0.38** | 0.46 | **0.50** | **0.52** | 0.54 | 0.56 | **0.57** | 0.59 | 0.60 |
| CT | 0.21 | **0.38** | 0.46 | **0.50** | 0.53 | 0.54 | **0.55** | 0.57 | **0.58** | 0.59 |
| CT + SGA + RTM | 0.21 | **0.38** | **0.44** | **0.50** | **0.52** | **0.52** | 0.56 | **0.57** | **0.58** | **0.58** |

**Note:** The values in **blue** indicate lower RMSE for CRN-based models, and values in **violet** indicate lower RMSE for CT-based models.

## 6.2 EXPERIMENTS WITH SEMI-SYNTHETIC DATA

To further validate our proposed methods, SGA and RTM, we conduct experiments on a semi-synthetic dataset based on real-world medical data from intensive care units. This dataset is generated following the approach of Melnychuk et al. (2022), which builds upon the MIMIC-III dataset (Johnson et al., 2016) to simulate patient trajectories with outcomes that reflect both endogenous and exogenous dependencies while incorporating treatment effects. Detailed information on dataset generation and hyperparameter settings is provided in Appendix F.2.

**Results and Analysis.** As shown in Table 3, applying SGA and RTM on top of CT and CRN *improves their performance compared to the vanilla models*. Furthermore, our methods yield performance *comparable to the SOTA models*. This is consistent with the findings reported in Melnychuk et al. (2022) that confounding may not be the primary challenge in this task, as there is also minimal difference between the performance of CT and CT ($\alpha$). This observation implies that, in this semi-synthetic dataset, *the level of confounding may be relatively low*. To this end, given that *the strength of our approach lies in reducing confounding*, it is expected that the performance gain is marginal compared to existing state-of-the-art methods.

## 7 CONCLUSION

In this work, we introduce two novel techniques—Sub-treatment Group Alignment (SGA) and Random Temporal Masking (RTM)—to enhance counterfactual outcome estimation in time series. SGA addresses time-varying confounding by aligning sub-treatment group distributions in the latent space, leading to tighter counterfactual error bound and more effective deconfounding, as supported by our theoretical analysis. RTM improves model robustness and generalization by encouraging focus on causal relationships through the random masking of covariates over time.

Our methods are flexible and can be integrated into various architectures, as we have demonstrated in our experiments that incorporating them into SOTA models like CRN and CT improve their performance. Experiments on fully synthetic and semi-synthetic datasets showed that combining SGA and RTM achieves superior performance, outperforming existing methods. Individually, each technique also contributes to performance improvements, highlighting their respective effectiveness.

**REPRODUCIBILITY STATEMENT.** We include rigorous definitions and complete proofs of our theoretical analysis in Appendix D. The code required to replicate all experiments is included in the supplementary materials, attached with the submission. Detailed descriptions of the experiments are located in Appendix F.1 for the fully synthetic dataset and Appendix F.2 for the semi-synthetic dataset. The hyperparameters necessary for reproducing our results are also included in these sections. Benchmark method performance is sourced from the GitHub repository of Melnychuk et al. (2022). The MIMIC-III dataset, used in our semi-synthetic experiment, is available for free download from the MIMIC-III Clinical Database Demo (version 1.4) on PhysioNet, licensed under the Open Data Commons Open Database License v1.0. All experiments were conducted on a computer cluster with A100-SXM4-40GB GPUs.

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

# A   POTENTIAL OUTCOMES FRAMEWORK WITH TIME-VARYING TREATMENTS AND OUTCOMES

Building on the potential outcomes framework (Rosenbaum & Rubin, 1983; Rubin, 2005), we extend these assumptions to accommodate time-varying treatments and outcomes, following Robins & Hernan (2008).

**Assumption A.1.** (Consistency) If $\bar{\mathbf{A}}_t = \bar{\mathbf{a}}_t$ is a given sequence of treatments for some patient, then $\mathbf{Y}_{t+1}[\bar{\mathbf{a}}_t] = \mathbf{Y}_{t+1}$ This means an individual's potential outcome under the observed treatment history is the observed outcome.

**Assumption A.2.** (Sequential Positivity) Positivity states that there is non-zero probability or not receiving any of the counterfactual treatment. It can be expressed as $0 \leq P(\mathbf{A}_t = \mathbf{a}_t | \bar{\mathbf{H}}_t = \bar{\mathbf{h}}_t) \leq 1$, if $P(\bar{\mathbf{H}}_t = \bar{\mathbf{h}}_t) > 0$.

**Assumption A.3.** (Sequential Ignorability) There is no unobserved confounding of treatment at any time and any future outcome. This can be expressed as $\mathbf{A}_t \perp\!\!\!\perp \mathbf{Y}_{t+1}[\mathbf{a}_t] | \bar{\mathbf{H}}_t, \forall\, \mathbf{a}_t \in \mathcal{A}$.

Using assumptions A.1–A.3, Robins (1986) establishes the sufficient conditions for identifiability through G-computation, ensuring that causal effects can be appropriately identified.

# B   CAUSAL GRAPH

Fig 4 visualizes Causal Directed Acyclic Graphs (DAGs) illustrating causal relationships. In the static (non-time-series) scenario, we have $A$ as the treatment assignment, $X$ as the covariate, and $Y$ as the outcome. In the time-series scenario, $T$ is the treatment sequence, $X_t \cup V$ represents observed covariates at time $t$), and $Y_t$ is the outcome at time $t$. Here, $V$ denotes static covariates that do not change over time. The diagrams capture the dynamics of treatment effects over time, showing how each component influences subsequent outcomes within the causal framework.

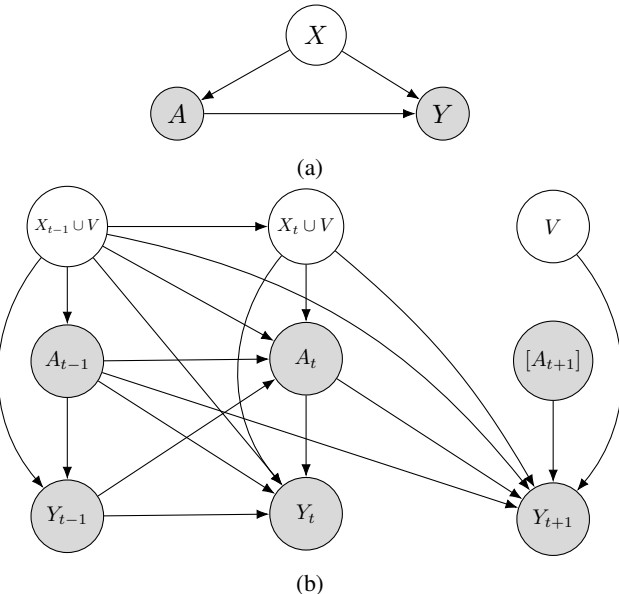

Figure 4: Causal Directed Acyclic Graphs (DAGs) Illustrating Causal Relationships. (a) demonstrate a static (non-time-series) scenario. (b) illustrates a time-series scenario.

# C   RELATED WORK

**Estimating counterfactual outcomes under static scenarios.** Many methods have been proposed to learn a *balanced* representation that aligns the distributions across various treatment groups, ef-

fectively addressing confounding in static settings. A foundational work in this area, CFRNet introduced by Shalit et al. (2017), establishes a generalization-error bound illustrating that the expected error in estimating individual treatment effects (ITE) is bounded by the sum of its standard generalization error and the discrepancy between the treated and control distributions induced by the representation. This concept has been further explored in several subsequent studies on deep causal inference (Yao et al., 2018; Kallus, 2020; Du et al., 2021; Jiang et al., 2023a;b). However, these methods primarily focus on binary treatments and static data, and their approach of aligning overall treated and control group distributions may not sufficiently adaptable to time-series data (Hernán et al., 2000; Mansournia et al., 2012), where time-dependent confounders make it difficult to disentangle the true effect of a treatment from these caused by the confounding variables.

**Estimating counterfactual outcomes over time.** Estimating counterfactual outcomes in time-series data is challenging due to time-varying confounders. Traditional methods such as G-computation and marginal structural models (Robins, 1986; Robins et al., 2000; Hernán et al., 2001; Robins & Hernan, 2008; Xu et al., 2016) often lack flexibility for complex datasets and rely on strong assumptions. To address these limitations, researchers have developed models that build on the potential outcomes framework initially proposed by Rubin (Rubin, 1978) and extended to time series by Robins & Hernan (2008). Notable among recent methods are Recurrent Marginal Structural Networks (RMSNs) (Lim, 2018), G-Net (Li et al., 2020), Counterfactual Recurrent Networks (CRN) (Bica et al., 2020), and the Causal Transformer (CT) (Melnychuk et al., 2022), which use approaches such as propensity networks and adversarial learning to mitigate the effects of time-varying confounding. The CRN employs recurrent neural networks like LSTMs, while the CT uses Transformer-based architectures, representing the state-of-the-art in this domain. However, practical challenges with adversarial training can affect the stability of causal effect estimations. Specifically, training adversarial networks can be challenging due to issues such as mode collapse and oscillations (Liang et al., 2018). Additionally, adversarial training minimizes the Jensen-Shannon divergence (JSD) only when the discriminator is optimal (Arjovsky & Bottou, 2017), which may not always be achievable in practice; even when the discrminator is optimal, using JSD optimizing relatively loose upper bounds on the counterfactual error. To address these challenges, we propose using the Wasserstein-1 distance. The Wasserstein distance is bounded above by the Kullback-Leibler divergence (JSD is a symmetrized and smoothed version of the Kullback–Leibler divergence) and provides stronger theoretical guarantees (Redko et al., 2017; Mansour et al., 2012). Moreover, the Wasserstein distance has stable gradients even when the compared distributions are far apart (Gulrajani et al., 2017), which enhances training stability and effectiveness.

**Masked language modeling.** Masked language modeling (MLM) is a common self-supervised pre-training technique for large language models. It operates by randomly masking certain words or tokens in the input, with the model trained to predict the masked tokens. BERT (Devlin, 2018) is the most well-known model that employs this technique. Recent studies have also demonstrated the effectiveness of MLM in enhancing generalization across various sequence-based tasks. For example, Chaudhary et al. (2020) showed that when combined with cross-lingual dictionaries, MLM not only improves predictions for the original masked word but also generalizes to its cross-lingual synonyms. Additionally, Czinczoll et al. (2024) illustrated how MLM enhances generalization in long-document tasks by leveraging higher-level semantic representations. Inspired by the success of masking strategies in language models, we introduce Random Temporal Masking (RTM) for time-series data. Unlike MLM, which focuses on predicting the masked inputs, RTM encourages the model to focus on information that is crucial not only for the current time point but also for future time points, preserve causal information, and reduce the risk of overfitting to factual outcomes.

## D  THEOREMS AND PROOFS

**Definition D.1** (Definition A4 in Shalit et al. (2017)). Let $\Phi : \mathcal{X} \rightarrow \mathcal{R}$ be a representation function. Let $h : \mathcal{R} \times \{0, 1\} \rightarrow \mathcal{Y}$ be an hypothesis defined over the representation space $\mathcal{R}$. The expected loss for the unit and treatment pair $(x, t)$ is:

$$\ell_{h,\Phi}(x, t) = \int_{\mathcal{Y}} L(Y_t, h(\Phi(x), t)) p(Y_t|x) dY_t,$$

where $L(\cdot, \cdot)$ is a loss function, from $\mathcal{Y} \times \mathcal{Y}$ to $\mathbb{R}_+$.

**Definition D.2** (Definition A5 in Shalit et al. (2017))**.** The expected factual loss and counterfactual losses of $h$ and $\Phi$ are, respectively:

$$\epsilon_F(h, \Phi) = \int_{\mathcal{X} \times \{0,1\}} \ell_{h,\Phi}(x, t)p(x, t)dxdt,$$

$$\epsilon_{CF}(h, \Phi) = \int_{\mathcal{X} \times \{0,1\}} \ell_{h,\Phi}(x, t)p(x, 1-t)dxdt,$$

where $p(x, t)$ is distribution over $\mathcal{X} \times \{0, 1\}$

**Definition D.3.** For some $K \geq 0$, the set of $K$-Lipschitz functions denotes the set of functions $f$ that verify:
$$\|f(x) - f(x')\| \leq K\|x - x'\|, \ \forall x, x' \in \mathcal{X}$$

Here, we assume that the hypothesis class $\mathbb{H}$ is a subset of $\lambda_H$-Lipschitz functions, where $\lambda_H$ is a positive constant, and we assume that the true labeling functions are $\lambda$-Lipschitz for some positive real number $\lambda$. Also if $f$ is differentiable, then a sufficient condition for $K$-Lipschitz constant is if $\|\frac{\partial f}{\partial x}\| \leq x$ for all $s \in \mathcal{X}$.

**Assumption D.4** (Assumption A2 in Shalit et al. (2017))**.** There exists a constant $K > 0$ such that for all $x \in \mathcal{X}, t \in \{0, 1\}, \|\frac{\partial p(Y_t|x)}{\partial x}\| \leq K$.

**Assumption D.5** (Assumption A3 in Shalit et al. (2017))**.** The loss function $L$ is differentiable, and there exists a constant $K_L > 0$ such that $\left|\frac{dL(y_1, y_2)}{dy_i}\right| \leq K_L$ for $i = 1, 2$. Additionally, there exists a constant $M$ such that for all $y_2 \in \mathcal{Y}, M \geq \int_{\mathcal{Y}} L(y_1, y_2)dy_1$.

**Definition D.6** (Definition A12 in Shalit et al. (2017))**.** Let $\frac{\partial \Phi(x)}{\partial x}$ be the Jacobian matrix of $\Phi$ at point $x$, i.e., the matrix of the partial derivatives of $\Phi$. Let $\sigma_{\max}(A)$ and $\sigma_{\min}(A)$ denote respectively the largest and smallest singular values of a matrix $A$. Define $\rho(\Phi) = \frac{\sup_{x \in \mathcal{X}} \sigma_{\max}(\frac{\partial \Phi(x)}{\partial x})}{\sigma_{\min}(\frac{\partial \Phi(x)}{\partial x})}$.

**Definition D.7** (Definition A13 in Shalit et al. (2017))**.** We will call a representation function $\Phi : \mathcal{X} \to \mathcal{R}$ Jacobian-normalized if $\sup_{x \in \mathcal{X}} \sigma_{\max}(\frac{\partial \Phi(x)}{\partial x}) = 1$

Note that any non-constant representation function $\Phi$ can be Jacobian-normalized by a simple scalar multiplication.

**Lemma D.8** (Lemma A3 in Shalit et al. (2017))**.** *Let $u = p(t = 1)$, then we have,*
$$\epsilon_F(h, \Phi) = u \cdot \epsilon_F^{t=1}(h, \Phi) + (1-u) \cdot \epsilon_F^{t=0}(h, \Phi)$$
$$\epsilon_{CF}(h, \Phi) = (1-u) \cdot \epsilon_{CF}^{t=1}(h, \Phi) + u \cdot \epsilon_{CF}^{t=0}(h, \Phi)$$

**Definition D.9.** Let $u = p(t = 1)$ be the marginal probability of treatment and assume $0 < u < 1$.
$$\epsilon_F^{\star}(h, \Phi) = (1-u)\epsilon_F^{t=1}(h, \Phi) + u\epsilon_F^{t=0}(h, \Phi)$$

Now using the Definition D.9, we rewrite Lemma A8 from Shalit et al. (2017). Then we get:

**Theorem D.10** (Lemma A8 from Shalit et al. (2017))**.** *Let $u = p(t = 1)$ be the marginal probability of treatment and assume $0 < u < 1$. Let $\Phi : \mathcal{X} \to \mathcal{R}$ be a one-to-one and Jacobian-normalized representation function. Let $K$ be the Lipschitz constant of the functions $p(Y_t|x)$ on $\mathcal{X}$. Let $K_L$ be the Lipschitz constant of the loss function $L$, and $M$ be as in Assumption D.5. Let $h : R \times \{0, 1\} \to Y$ be an hypothesis with Lipschitz constant $bK$:*
$$\epsilon_{CF}(h, \Phi) \leq \epsilon_F^{\star}(h, \Phi) + 2\left(M\rho(\Phi) + b\right) \cdot K \cdot K_L \cdot W_1(p_\Phi^0, p_\Phi^1), \tag{4}$$
*where $B_\Phi = (M\rho(\Phi) + b) \cdot K \cdot K_L$ is a constant and $p_\Phi^a$ is the distribution of the random variable $\Phi(X)$ conditioned on $A = a$, that is, representations for individuals receiving treatment $a \in \{0, 1\}$.*

**Definition D.11. Wasserstein Distance.** The Kantorovich-Rubenstein dual representation of the Wasserstein-1 distance (Villani, 2009) between two distributions $p_\Phi^0$ and $p_\Phi^1$ is defined as

$$W_1(p_\Phi^0, p_\Phi^1) = \sup_{\|f\|_L \le 1} \mathbb{E}_{x \sim p_\Phi^0}[f(x)] - \mathbb{E}_{x \sim p_\Phi^1}[f(x)],$$

where the supremum is over the set of 1-Lipschitz functions (all Lipschitz functions $f$ with Lipschitz constant $L \le 1$. For notational simplicity, we use $D(X_1, X_2)$ to denote a distance between the distributions of any pair of random variables $X_1$ and $X_2$. For instance, $W_1(\Phi(X_0), \Phi(X_1))$ denotes the Wasserstein-1 distance between the distributions of the random variables $\Phi(X_0)$ and $\Phi(X_1)$ for any transformation $\Phi$.

Next, motivated by the generalization bound in the field of domain adaptation Liu et al. (2023), we prove that sub-treatment group alignment ***improves the original alignment method in Theorem 4.1 by optimizing a tighter upper bound of the counterfactual error***.

**Definition D.12.** (Wasserstein-like distance between Gaussian Mixture Models) Assume that both $X_0$ and $X_1$ are mixtures of $K$ sub-domains. In other words, we have $p_\Phi^0 = \sum_{k=1}^K w_k^0 P_{\Phi,k}^0$ and $p_\Phi^1 = \sum_{k=1}^K w_k^1 P_{\Phi,k}^1$ where for $a \in 0, 1$, $w_k^a$ represents the proportion of the $k$-th sub-distribution in treatment group $a$. $P_{\Phi,k}^a$ denotes the distribution of the representations in the $k$-th sub-group under treatment $a$. We define:

$$MW_1(p_\Phi^0, p_\Phi^1) = \min_{w \in \Pi(\mathbf{w^0}, \mathbf{w^1})} \sum_{k=1}^K \sum_{k'=1}^K w_{k,k'} W_1(P_{\Phi,k}^0, P_{\Phi,k'}^1) \tag{5}$$

where $\mathbf{w^0} \doteq [w_1^0, \ldots, w_K^0]$ and $\mathbf{w^1} \doteq [w_1^1, \ldots, w_K^1]$ belong to $\Delta^K$ (the $K-1$ probability simplex). $\Pi(w^0, w^1)$ represents the simplex $\Delta^{K \times K}$ with marginals $\mathbf{w^0}$ and $\mathbf{w^1}$.

**Lemma D.13** (Extension to Lemma 4.1 of Delon & Desolneux (2020)). *Let $\mu_0 = \sum_{k=1}^{K_0} \pi_0^k \mu_0^k$ with $\mu_0^k = \mathcal{N}(m_0^k, \Sigma_0^k)$ and $\mu_1 = \sum_{k=1}^{K_1} \pi_1^k \delta_{m_1^k}$. Let $\tilde{\mu}_0 = \sum_{k=1}^{K_0} \pi_0^k \delta_{m_0^k}$ ($\tilde{\mu}_0$ only retains the means of $\mu_0$). Then,*

$$MW_1(\mu_0, \mu_1) \le W_1(\tilde{\mu}_0, \mu_1) + \sum_{k=1}^{K_0} \pi_0^k \sqrt{tr\left(\Sigma_0^k\right)}$$

*where $\pi_{\mathbf{0}} \doteq [\pi_0^1, \ldots, \pi_0^k]$ and $\pi_{\mathbf{1}} \doteq [\pi_1^1, \ldots, \pi_1^k]$ belong to $\Delta^K$ (the $K-1$ probability simplex)*

*Proof.*

$$
\begin{aligned}
MW_1(\mu_0, \mu_1) &= \inf_{w \in \Pi(\pi_0, \pi_1)} \sum_{k,l} w_{k,l} W_1(\mu_0^k, \delta_{m_1^l}) \\
&\le \inf_{w \in \Pi(\pi_{\mathbf{0}}, \pi_{\mathbf{1}})} \sum_{k,l} w_{k,l} W_2(\mu_0^k, \delta_{m_1^l}) \\
&= \inf_{w \in \Pi(\pi_{\mathbf{0}}, \pi_{\mathbf{1}})} \sum_{k,l} w_{k,l} \left[ \sqrt{||m_1^l - m_0^k||^2 + \text{tr}\left(\Sigma_0^k\right)} \right] \\
&\le \inf_{w \in \Pi(\pi_{\mathbf{0}}, \pi_{\mathbf{1}})} \sum_{k,l} w_{k,l} ||m_1^l - m_0^k|| + \sum_k \pi_0^k \sqrt{\text{tr}\left(\Sigma_0^k\right)} \\
&\le W_1(\tilde{\mu}_0, \mu_1) + \sum_{k=1}^{K_0} \pi_0^k \sqrt{\text{tr}\left(\Sigma_0^k\right)}
\end{aligned}
\tag{6}
$$

$\square$

*Remark* D.14. We use $\mu_0$, $\mu_1$, and $\tilde{\mu}_0$ to represent a general scenario for measuring the distance between a Gaussian mixture and a mixture of Diract distributions. In the following proofs, we will utilize the defined notation. For instance, $\mu_0$ can be denoted as $p_\Phi^0$, while $\tilde{\mu}_0$ corresponds to $\tilde{p_\Phi^0}$.

**Theorem D.15** (Extension to Proposition 6 in (Delon & Desolneux, 2020))**.** *Let $p_\Phi^0$ and $p_\Phi^1$ be two Gaussian mixtures with $p_\Phi^0 = \sum_{k=1}^{K} w_k^0 P_{\Phi,k}^0$ and $p_\Phi^1 = \sum_{k=1}^{K} w_k^1 P_{\Phi,k}^1$. For all $k$, $P_{\Phi,k}^0 / P_{\Phi,k}^1$ are Gaussian distributions with mean $m_k^0 / m_k^1$ and covariance $\Sigma_k^0 / \Sigma_k^1$. If for $\forall$ $k$, $k'$, we assume there exists a small constant $\epsilon > 0$, such that $\max_k(trace(\Sigma_k^0)) \leq \epsilon$ and $\max_{k'}(trace(\Sigma_{k'}^1)) \leq \epsilon$. then:*

$$MW_1(p_\Phi^0, p_\Phi^1) \leq W_1(p_\Phi^0, p_\Phi^1) + 4\sqrt{\epsilon} \tag{7}$$

*Proof.* Here, we follow the same structure of the proof for Wassertein-2 in Delon & Desolneux (2020). Let $(P_\phi^0)_n$ and $((P_\phi^1)_n$ be two sequences of mixtures of Dirac masses respectively converging to $P_\phi^0$ and $P_\phi^1$ in $\mathcal{P}_1(\mathbb{R}^d)$. Since $MW_1$ is a distance,

$$MW_1(P_\phi^0, P_\phi^1) \leq MW_1((P_\phi^0)^n, (P_\phi^1)^n) + MW_1(P_\phi^0, (P_\phi^0)^n) + MW_1(P_\phi^1, (P_\phi^1)^n)$$
$$= W_1((P_\phi^0)^n, (P_\phi^1)^n) + MW_1(P_\phi^0, (P_\phi^0)^n) + MW_1(P_\phi^1, (P_\phi^1)^n)$$

We can study the limits of these three terms when n $\rightarrow +\infty$

First, observe that $MW_1(P_\phi^0, P_\phi^1) = W_1((P_\phi^0)^n, (P_\phi^1)^n) \underset{n\to+\infty}{\rightarrow} W_1(P_\phi^0, P_\phi^1)$ since $W_1$ is continuous on $\mathcal{P}_1(\mathbb{R}^d)$.

Second, based on Lemma D.13, we have that

$$MW_1(P_\phi^0, (P_\phi^0)^n) \leq W_1(\tilde{P_\phi^0}, (P_\phi^0)^n) + \sum_{k=1}^{K} w_k^0 \sqrt{\text{tr}(\Sigma_k^0)} \underset{n\to+\infty}{\rightarrow} W_1(\tilde{P_\phi^0}, P_\phi^0) + \sum_{k=1}^{K} w_k^0 \sqrt{\text{tr}(\Sigma_k^0)}$$

We observe that $x \mapsto \sqrt{x}$ is a concave function, thus by Jensen's inequality, we have that

$$\sum_{k=1}^{K} w_k^0 \sqrt{\text{tr}(\Sigma_k^0)} \leq \sqrt{\sum_{k=1}^{K} w_k^0 \text{tr}(\Sigma_k^0)}$$

Also By Jensen's inequality, we have that,

$$W_1(\tilde{P_\phi^0}, P_\phi^0) \leq W_2(\tilde{P_\phi^0}, P_\phi^0).$$

And from Proposition 6 in (Delon & Desolneux, 2020), we have

$$W_2(\tilde{P_\phi^0}, P_\phi^0) \leq \sqrt{\sum_{k=1}^{K} w_k^0 \text{tr}(\Sigma_k^0)}$$

Similarly for $MW_1(P_\phi^1, (P_\phi^1)^n)$ the same argument holds. Therefore we have,

$$\lim_{n\to\infty} MW_1(P_\phi^0, (P_\phi^0)^n) \leq 2\sqrt{\sum_{k=1}^{K} w_k^0 \text{tr}(\Sigma_k^0)}$$

And

$$\lim_{n\to\infty} MW_1(P_\phi^1, (P_\phi^1)^n) \leq 2\sqrt{\sum_{k=1}^{K} w_k^1 \text{tr}(\Sigma_k^1)}$$

We can conclude that:

$$MW_1(P_\phi^0, P_\phi^1) \leq \lim_{n\to\infty} \inf (W_1((P_\phi^0)^n, (P_\phi^1)^n) + MW_1(P_\phi^0, (P_\phi^0)^n) + MW_1(P_\phi^1, (P_\phi^1)^n))$$

$$\leq W_1(P_\phi^0, P_\phi^1) + 2\sqrt{\sum_{k=1}^{K} w_k^0 \text{tr}(\Sigma_k^0)} + 2\sqrt{\sum_{k=1}^{K} w_k^1 \text{tr}(\Sigma_k^1)}$$

$$\leq W_1(P_\phi^0, P_\phi^1) + 4\sqrt{\epsilon}$$

This concludes the proof. $\square$

**Theorem D.16** (SGA Improves Generalization Bounds). *Under the following assumptions: **A1.** For all $k$, the sub-distributions $P_{\Phi,k}^0$ and $P_{\Phi,k}^1$ are Gaussian distributions with means $m_k^0$ and $m_k^1$, and covariances $\Sigma_k^0$ and $\Sigma_k^1$, respectively. The distance between corresponding sub-distributions is less than or equal to the distance between non-corresponding sub-distributions, i.e., $W_1(P_{\Phi,k}^0, P_{\Phi,k}^1) \leq W_1(P_{\Phi,k}^0, P_{\Phi,k'}^1)$ for $k \neq k'$.*

***A2.** There exists a small constant $\epsilon > 0$, such that $\max\limits_{1 \leq k \leq K}(tr(\Sigma_k^0)) \leq \epsilon$ and $\max\limits_{1 \leq k \leq K}(tr(\Sigma_k^1)) \leq \epsilon$. Then the following inequalities hold:*

$$\sum_{k=1}^K w_k^1 W_1(P_{\Phi,k}^0, P_{\Phi,k}^1) \leq W_1(p_\Phi^0, p_\Phi^1) + \delta_c,$$

*where $\delta_c$ is $4\sqrt{\epsilon}$.*

This theorem shows that the weighted sum of the Wasserstein distances between the aligned sub-treatment groups (as performed by SGA) is bounded above by the Wasserstein distance between the overall treatment and control distributions, plus a small constant $\delta_c$. Therefore, by minimizing the sum of distances between corresponding sub-groups, SGA effectively tightens the generalization bound compared to methods that align the overall distributions.

*Proof.* Let $\mathbf{w^0} \doteq [w_1^0, \ldots, w_K^0]$ and $\mathbf{w^1} \doteq [w_1^1, \ldots, w_K^1]$ belong to $\Delta^K$ (the $K - 1$ probability simplex). $\Pi(w^0, w^1)$ represents the simplex $\Delta^{K \times K}$ with marginals $\mathbf{w^0}$ and $\mathbf{w^1}$. For any $w \in \Pi(w^0, w^1)$, we can express $w_k^1 = \sum_{k'=1}^K w_{k,k'}$. Based on assumption A1, we have:

$$\sum_{k=1}^K w_k^1 W_1(P_{\Phi,k}^0, P_{\Phi,k}^1) = \sum_{k=1}^K \sum_{k'=1}^K w_{k,k'} W_1(P_{\Phi,k}^0, P_{\Phi,k}^1)$$
$$\leq \sum_{k=1}^K \sum_{k'=1}^K w_{k,k'} W_1(P_{\Phi,k}^0, P_{\Phi,k'}^1).$$

Thus, we have (with $MW_1(p_\Phi^0, p_\Phi^1)$ defined in Appendix D.12):

$$\sum_{k=1}^K w_k^1 W_1(P_{\Phi,k}^0, P_{\Phi,k}^1) \leq \min_{w \in \Pi(\mathbf{w^0}, \mathbf{w^1})} \sum_{k=1}^K \sum_{k'=1}^K w_{k,k'} W_1(P_{\Phi,k}^0, P_{\Phi,k'}^1) \tag{8}$$
$$= MW_1(p_\Phi^0, p_\Phi^1).$$

From Theorem D.15, we have:

$$MW_1(p_\Phi^0, p_\Phi^1) \leq W_1(p_\Phi^0, p_\Phi^1) + 4\sqrt{\epsilon}. \tag{9}$$

Combining the above results:

$$\sum_{k=1}^K w_k^1 W_1(P_{\Phi,k}^0, P_{\Phi,k}^1) \leq W_1(p_\Phi^0, p_\Phi^1) + 4\sqrt{\epsilon}. \tag{10}$$

$\square$

With Theorem 4.2, we demonstrate that aligning sub-treatment groups via SGA leads to a tighter bound on the counterfactual loss at each time step. Specifically, the new generalization bound incorporates the weighted sum of distances between corresponding sub-groups, which SGA minimizes through targeted alignment.

# E  IMPLEMENTATION DETAILS AND ALGORITHM

**Computing the Uniform Mixture of Sub-treatment Groups**  In our implementation of the SGA loss, for each time step $t$ and each cluster $k$, we compute the uniform mixture of sub-treatment groups $\phi_E^{t,k}$.
To compute this uniform mixture, we perform the following steps:

1. **Concatenate representations across treatments:**
   For the $k$-th cluster at time $t$, we collect the representations from all treatment groups:

   $$\phi_E^{t,k}(\mathbf{H}_t) = \bigcup_{a \in \mathcal{A}} \phi_E^{t,a,k}(\mathbf{H}_t),$$

   where $\phi_E^{t,a,k}(\mathbf{H}_t)$ denotes the representations of samples in the $k$-th sub-group of treatment $a$ at time $t$.

2. **Shuffle and subsample:**
   We shuffle the concatenated representations to ensure that samples from different treatments are thoroughly mixed. Then we select $\frac{1}{|\mathcal{A}|}$ from the concatenated representations as $\phi_E^{t,k}$.

---

**Algorithm 1** Counterfactual Outcome Estimation with Sub-treatment Group Alignment (SGA) and Random Temporal Masking (RTM)

---

**Require:**

    $\mathcal{D} = \{(\mathbf{X}_i^t, \mathbf{A}_i^t, \mathbf{Y}_i^{t+1})\}_{i=1}^N$: Training data for $N$ individuals for t = 1,...,T

    $\theta_E, \theta_Y$: Parameters of encoder $\phi_E$ and regressor $f_Y$

    $\lambda$: Hyperparameter for $L_D$

    $K$: Number of sub-treatment groups (clusters)

    $\mathcal{A}$: Set of possible treatments

    MaskProb: Probability of masking covariates in RTM

    $\eta$: Learning rate

    $\ell(\cdot, \cdot)$: Loss function (e.g., mean squared error)

1: **Apply Random Temporal Masking (RTM):**
2: **for** each time step $t = 1$ to $T$ **do**
3:     With probability MaskProb, replace $X^t$ with Gaussian noise
4: **end for**
5: Initialize $L_Y = 0, L_D = 0$
6: **for** each time step $t = 1$ to $T$ **do**
7:     $\Phi_E(\mathbf{H}_t) = \phi_E(\mathbf{H}_t, A^t)$
8:     $\hat{Y} = f_Y(\Phi_E(\mathbf{H}_t))$
9:     **Compute Factual Outcome Loss:**
10:     $L_Y = L_Y + \ell(Y^{t+1}, \hat{Y}^{t+1})$
11:     **Compute SGA Loss:**
12:     **for** each treatment $a \in \mathcal{A}$ **do**
13:         **Cluster representations into $K$ sub-groups:**
14:         Apply GMM to $\Phi_E(\mathbf{H}_t)$ to obtain clusters $\{\phi_E^{t,a,k}\}_{k=1}^K$
15:         Compute weights $w_k^{t,a} = \frac{n_k^{t,a}}{n^{t,a}}$, where $n_k^{t,a}$ is the number of samples in cluster $k$, $n^{t,a}$ is the total number of samples with treatment $a$ at time $t$
16:     **end for**
17:     **Compute uniform mixture of sub-groups $\phi_E^{t,k}$**
18:     **Compute SGA loss at time $t$:**
19:     $L_D = L_D + \sum_{k=1}^K \sum_{a \in \mathcal{A}} w_k^{t,a} \cdot W_1\left(\phi_E^{t,a,k}, \phi_E^{t,k}\right)$
20:     **Compute Total Loss:**
21:     $L = L_Y + \lambda L_D$
22:     **Update model parameters:**
23:     $\theta_E \leftarrow \theta_E - \eta \nabla_{\theta_E} L$
24:     $\theta_Y \leftarrow \theta_Y - \eta \nabla_{\theta_Y} L$
25: **end for**

---

# F    EXPERIMENTS

## F.1    FULLY SYNTHETIC DATASET

### F.1.1    DATASET GENERATION

Dataset generation follows the identical setup as Bica et al. (2020); Melnychuk et al. (2022). The tumor growth simulator (Geng et al., 2017) models the tumor volume $Y_{t+1}$ after $t + 1$ days of diagnosis. It includes two binary treatments: (i) radiotherapy $A_t^r$ and (ii) chemotherapy $A_t^c$. These treatments influence tumor progression as follows:

- **Radiotherapy** has an immediate impact, denoted by $d(t)$, on the tumor volume at the next time step.
- **Chemotherapy** impacts future tumor progression with an exponentially decaying effect $C(t)$.

The model is described by the equation:

$$Y_{t+1} = \left( 1 + \rho \log \left( \frac{K}{Y_t} \right) - \beta_c C_t - (\alpha_r d_t + \beta_r d_t^2) + \varepsilon_t \right) Y_t$$

where $\varepsilon_t \sim N(0, 0.01^2)$ is independent noise, and the variables $\beta_c, \alpha_r, \beta_r$ represent the response characteristics for each individual. These parameters are drawn from truncated normal distributions comprising three mixture components. For a full list of parameter values, the code implementation should be consulted.

Time-varying confounding is accounted for through biased treatment assignments, where treatment allocation is identical across both therapies $A_t^r$ and $A_t^c$:

$$A_t^r, A_t^c \sim \text{Bernoulli} \left( \sigma \left( \frac{\gamma}{D_{\max}} (\bar{D}_{15} \bar{Y}_{t-1} - \frac{D_{\max}}{2}) \right) \right)$$

In this formula, $\sigma(\cdot)$ represents the sigmoid function, $D_{\max}$ is the maximum tumor diameter in the last 15 days, and $\gamma$ is the confounding parameter. $\bar{D}_{15}(\bar{Y}_{t-1})$ refers to the average tumor diameter over the previous 15 days. If $\gamma = 0$, the treatment assignment is fully randomized, but for increasing values of $\gamma$, time-varying confounding gradually intensifies. More details can be found in Appendix J in CT Melnychuk et al. (2022).

### F.1.2    EXPERIMENTS SETUP

**One-step-ahead prediction.** To evaluate one-step-ahead predictions, we utilize the counterfactual trajectories simulated in CT. Our approach involves comparing our estimated outcomes $Y_{t+1}$ against all four possible combinations of one-step-ahead counterfactual outcomes. This effectively captures the tumor volumes under every possible treatment assignment at the next time step.

**$\tau$-step-ahead prediction.** For multi-step-ahead predictions, the number of potential outcomes for $Y_{t+2}, ..., Y_{t+\tau_{max}}$ grows exponentially with the prediction horizon $\tau_{max}$. To manage this complexity, and following the methodology in CT, we employ a single sliding treatment strategy. This approach is motivated by the importance of treatment timing in clinical settings. As discussed in the intro-duction, consider the treatment of *Ductal Carcinoma In Situ* (DCIS), where the timing of surgical intervention is critical: delaying surgery might allow the cancer to progress to an invasive stage, while performing it too early could lead to unnecessary invasiveness. To assess whether our models can identify the optimal timing for treatment, we simulate trajectories with a single treatment event that is iteratively shifted across a window ranging from time t to $t + \tau_{max} - 1$.

**Performance evaluation.** In line with Melnychuk et al. (2022), we evaluate model performance using the mean Root Mean Square Error (RMSE) on the test set, which consists of hold-out data. The RMSE is normalized by dividing by the maximum tumor volume $V_{max} = 1150 \text{cm}^3$. Additionally, we report the test RMSE calculated exclusively on the counterfactual outcomes following the rolling origin, thereby isolating the evaluation from historical factual patient trajectories.

### F.1.3 EMPIRICAL ANALYSIS OF OUR PROPOSED GENERALIZATION BOUND

As shown in Fig 5, here we empirically evaluate the proposed generalization bound. we provide empirical evidence that Sub-treatment Group Alignment (SGA) results in a much tighter upper bound compared to the original method in Theorem 4.1.

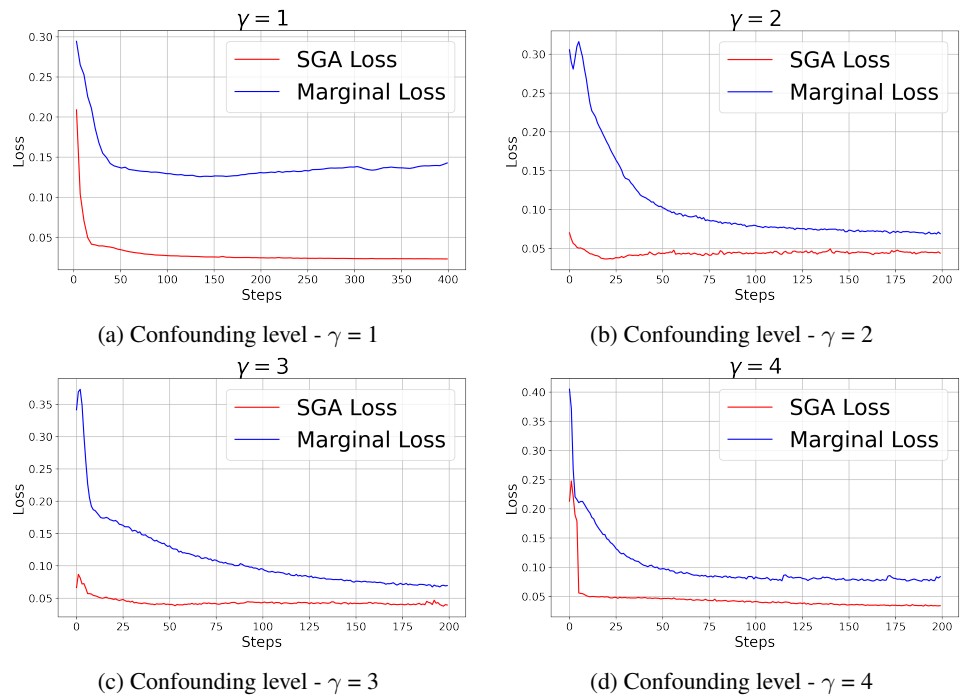

(a) Confounding level - $\gamma = 1$                     (b) Confounding level - $\gamma = 2$

(c) Confounding level - $\gamma = 3$                     (d) Confounding level - $\gamma = 4$

Figure 5: Empirical results for Sub-treatment Group Alignment (SGA) vs. the original method in Theorem 4.1 with varying confounding levels.

### F.1.4 ANALYSIS OF REPRESENTATION SPACE

We visualize the feature spaces learned by our Sub-treatment Group Alignment (SGA) method. As shown in Figure 6, SGA is able to learn treatment-invariant representations, which improves performance in counterfactual outcome estimation.

### F.1.5 MODEL HYPERPARAMETERS

Benchmark method hyperparameters and performance are sourced from the GitHub repository of Melnychuk et al. (2022).

Table 4: Model hyperparameters used for the fully-synthetic dataset.

| | $\gamma = 0$ | $\gamma = 1$ | $\gamma = 2$ | $\gamma = 3$ | $\gamma = 4$ |
|---|---|---|---|---|---|
| CT + SGA + RTM | batch size = 2048, learning_rate = 0.025, $\lambda$ = 0.0001, dropout rate = 0.2, Adam | batch size = 1024, learning_rate = 0.02, $\lambda$ = 0.0001, dropout rate = 0.1, Adam | batch size = 512, learning_rate = 0.02, $\lambda$ = 0.001, dropout rate = 0.1, Adam | batch size = 512, learning_rate = 0.03, $\lambda$ = 0.001, dropout rate = 0.1, Adam | batch size = 1024, learning_rate = 0.01, $\lambda$ = 0.001, dropout rate = 0.1, Adam |
| CRN + SGA + RTM | encoder batch size = 1024, encoder learning_rate = 0.005, encoder dropout rate = 0.1, decoder batch size = 4096, decoder learning_rate = 0.01, decoder dropout rate = 0.2 , $\lambda$ = 0.0001, Adam | encoder batch size = 1024, encoder learning_rate = 0.005, encoder dropout rate = 0.1, decoder batch size = 4096, decoder learning_rate = 0.01, decoder dropout rate = 0.1 , $\lambda$ = 0.0001, Adam | encoder batch size = 1024, encoder learning_rate = 0.005, encoder dropout rate = 0.2, decoder batch size = 4096, decoder learning_rate = 0.01, decoder dropout rate = 0.1 , $\lambda$ = 0.0001, Adam | encoder batch size = 1024, encoder learning_rate = 0.005, encoder dropout rate = 0.2, decoder batch size = 4096, decoder learning_rate = 0.01, decoder dropout rate = 0.1 , $\lambda$ = 0.001, Adam | encoder batch size = 1024, encoder learning_rate = 0.005, encoder dropout rate = 0.2, decoder batch size = 4096, decoder learning_rate = 0.01, decoder dropout rate = 0.1 , $\lambda$ = 0.01, Adam |

## F.2 SEMI-SYNTHETIC DATASET

We used the identical semi-synthetic dataset generated by Melnychuk et al. (2022), which is based on real-world medical data from intensive care units, to validate our model with high-dimensional,

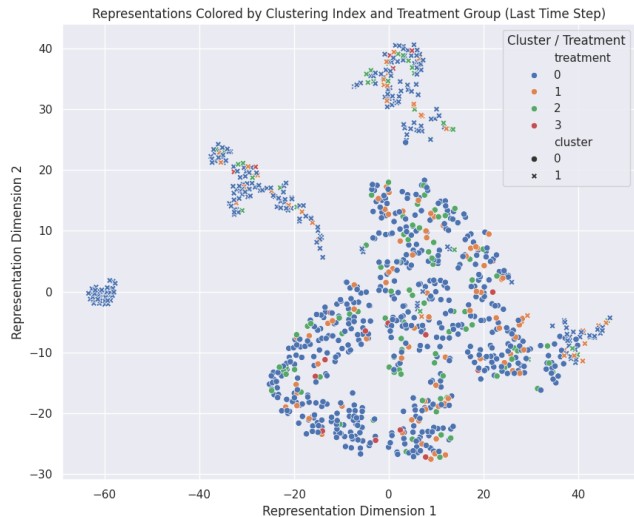

Figure 6: Representations at the last time point in training under high-confounding scenarios (i.e., $\gamma = 4$), with features projected to two dimensions using UMAP.

long-range patient trajectories. As outlined in Melnychuk et al. (2022), this dataset builds on the MIMIC-III dataset and simulates patient trajectories with both endogenous and exogenous dependencies, taking treatment effects into account (Johnson et al., 2016). This setup allows us to control for confounding in our experiments. The use of semi-synthetic data is important here, as real-world data lacks ground-truth counterfactuals, which are necessary for evaluating our methods' performances. To make our manuscript self-sustained, we hereby summarize the setup elaborated in Causal Transformer (Melnychuk et al., 2022). Full details on the data generation process can be found in Appendix K Melnychuk et al. (2022).

Following (Melnychuk et al., 2022), we utilized MIMIC-extract (Wang et al., 2020) based on the MIMIC-III dataset (Johnson et al., 2016). The data were preprocessed with forward and backward imputation for missing values and standardization of continuous features. Our dataset included 25 time-varying signals and 3 static covariates (gender, ethnicity, age), yielding 44 total features ($d_w = 44$) after one-hot-encoding.

The simulation follows four main steps:

1. **Cohort Selection**
   1,000 patients whose ICU stays lasted between 20 and 100 hours are sampled .

2. **Untreated Outcomes**
   For each patient $i$, simulated $d_y$ untreated outcomes $\mathbf{Z}_t^{j,(i)}$ are simulated by combining:

   - A B-spline term as an endogenous component
   - Random function $g^{j,(i)}(t)$
   - Exogenous covariate dependencies $f_Z^j(\mathbf{X_t}^{(i)})$
   - Independent Gaussian noise $\epsilon_t \sim N(0, 0.005^2)$

   $$\mathbf{Z}_t^{j,(i)} = \alpha_S^j \text{B-spline}(t) + \alpha_g^j g^{j,(i)}(t) + \alpha_f^j f_Z^j(\mathbf{X_t}^{(i)}) + \epsilon_t$$

3. **Treatment Assignment**
   We generated binary treatment indicators $\mathbf{A_t}^l$, $l = 1, ..., d_a$, based on previous outcomes and covariates, using a sigmoid function:

   $$p_{\mathbf{A_t}^l} = \sigma(\gamma_A^l \bar{A}_{T_l}(\bar{Y}_{t-1}) + \gamma_X^l f_Y^l(X_t) + b_l)$$

$$\mathbf{A_t}^l \sim \text{Bernoulli}(p_{\mathbf{A_t}^l})$$

Confounding is added by a subset of current time-varying covariates via a random function $f_Y^l(X_t)$, and $f_Y^l(\cdot)$ is sampled from an RFF approximation of a Gaussian process.

4. **Treatment Effects**

   In this step, treatments are applied to the initial untreated outcomes. We start by setting $\mathbf{Y_1} = \mathbf{Z_1}$, where each treatment $l$ influences an outcome $j$ with an immediate, maximum effect $\beta_{lj}$ after application. The treatment effect occurs within a time window from $t - w^l$ to $t$, with effect decreasing according to an inverse-square decay over time. The effect is also scaled by the treatment probability $p_{\mathbf{A_t^1}}$. When multiple treatments are involved, their combined effect is calculated by taking the minimum across all treatment impacts.

   The aggregated treatment effect is given by:

   $$E^j(t) = \sum_{i=t-w^l}^{t} \frac{\min_{l=1,\ldots,d_a} \mathbb{1}_{[\mathbf{A_i^l} = 1]} p_{\mathbf{A_i^1}} \beta_{lj}}{(w^l - i)^2}$$

5. **Combining Treatment Effects**

   We then add the simulated treatment effect $E^j(t)$ to the untreated outcome $Z_t^j$ to get the final outcome:

   $$Y_t^j = Z_t^j + E^j(t)$$

6. **Dataset Generation**

   The semi-synthetic dataset was generated using the above framework. For the exact parameter values used in the simulation, please refer to the GitHub repository of Melnychuk et al. (2022). Following the setup in CT, we used the simulated three synthetic binary treatments ($d_a = 3$) and two synthetic outcomes ($d_y = 2$). We also use the identical setup and split the 1000-patient cohort into training, validation, and test sets, with a 60%/20%/20% split. For one-step-ahead prediction, all $2^3 = 8$ counterfactual outcomes were simulated. For multiple-step-ahead prediction, we sampled 10 random trajectories for each patient and time step, with $\tau_{\max} = 10$.

### F.2.1 MODEL HYPERPARAMETERS

Benchmark method hyperparameters and performance are sourced from the GitHub repository of Melnychuk et al. (2022).

Table 5: Model hyperparameters used for the semi-synthetic dataset.

| | |
|---|---|
| CT + SGA + RTM | batch size = 64, learning_rate = 0.01, $\lambda = 0.0001$, dropout rate = 0.1, Adam |
| CRN + SGA + RTM | encoder batch size = 128, encoder learning_rate = 0.001, encoder dropout rate = 0.1, decoder batch size = 512, decoder learning_rate = 0.0001, decoder dropout rate = 0.1 , $\lambda = 0.0001$, Adam |

