# OpenReview forum: "Counterfactual Outcome Estimation in Time Series via Sub-treatment Group Alignment and Random Temporal Masking"
_ICLR.cc/2025/Conference — Submitted to ICLR 2025_

### Official Review · Reviewer_Qg2s · 2024-10-25

**Soundness:** 2
**Presentation:** 2
**Contribution:** 3
**Rating:** 5
**Confidence:** 4

**Summary:**

This paper provides two novel contributions that can be added to many current representation learning-based frameworks for counterfactual estimation on time series to tighten the upper bound and provide improved estimation: Sub-treatment Group Alignment (SGA) and Random Temporal Masking (RTM). SGA first identifies sub-treatment groups in the representation space through Gaussian Mixture Models, and subsequently aligns the corresponding sub-groups of different treatment groups. RTM uses random covariate masking, where the input covariates at randomly selected time points are replaced with Gaussian noise during training. Both methods are validated through comprehensive experiments on synthetic and semi-synthetic datasets.

**Strengths:**

Two methods are clearly described and some theoretical analysis is provided. The experimental results reported seem to support the method claim in the majority of setup, although they are tested only from one run.

**Weaknesses:**

The main weakness in this paper is its experiment validity. The robustness of the model to the random masking algorithm has not been evaluated. The main results in Tables 2 and 3 report only a single run. However, both the synthetic/semi-synthetic data-generating processes (DGP) in Appendix F and the random masking algorithm in Appendix E involve random sampling. It would be more convincing to repeat these experiments across multiple runs to assess the variability and robustness of the model's performance under different sampling conditions/mask ratios.

**Questions:**

1. From Figure 1 I could get some idea of subtreatment group alignment (say for T=1 in the representation space the treatment and control has very different distributions but in the sub-treatment group space they are very overlapped). Could you clarify how you get back from SGA to respretation space? What are the GMM coefs here?
2. This has been brought up in weakness. SGA/RTM improves existing architecture RSME in the majority of the setup, but will the results be still consistent in different runs?
3. What is the ratio of random mask in the experiments? Does it have an impact on performance?
4.  How are the number of sub-treatment group determined in your experiments, through some sort of cross validation? Could really not find them in Table 4/5 Appendix.

---

### Official Review · Reviewer_cHJk · 2024-10-31

**Soundness:** 2
**Presentation:** 2
**Contribution:** 2
**Rating:** 5
**Confidence:** 4

**Summary:**

Aiming at the unobservable counterfactual results and the complexity of confounding in time series, this manuscript proposes two new techniques: Subtreatment Group Alignment (SGA) and Random Temporal Masking (RTM). The SGA first identifies the sub-processing groups in the representation space by Gaussian Mixture Models (GMMs), and then sorts the corresponding sub-processing groups of the different processing groups. At an intuitive level, the arrangement of subgroups makes the arrangement of treatment groups more accurate and thus more effective in reducing confusion. RTM enhances the generalization ability of the model across time series when SGA deals with single time point mixing. Inspired by masking language modeling, RTM uses random covariates masking, and Gauss noise is used to replace the input covariates at randomly selected time points during training. Experiments on fully synthesized and semi-synthesized datasets show that both SGA and RTM can achieve better performance than existing methods.

**Strengths:**

1. The research method used in this manuscript is scientific, reasonable and innovative. The proof of the theorem is detailed and precise.
2. The experimental results show that SGA and RTM have good performance.
3. This manuscript has a strict logical structure and accurately conveys the author's research intention and findings, enabling readers to easily understand the contents of the paper.

**Weaknesses:**

1. The theorem proving for RTM is not enough.
2. We suggest adding the results of SGA+RTM to Table 1 and Table 2, and the results of +SGA and +RTM to Table 3, to compare the performance difference between using both SGA and RTM and using SGA or RTM alone.
3. In Figure 3(b) , the Normalized RMSE of RMSNs is similar to CT+SGA+/RTM and CRN+SGA +/RTM when γ=4. We suggest adding the results when γ is higher, and explain why you get such results.
4. This manuscript does not provide any direction on future research or the next steps following this study's completion. Please consider providing some guidance in this area.

**Questions:**

1. Why choose to apply SGA and RTM to CRN and CT instead of other benchmarking methods?
2. Why choose GMM over other clustering algorithms in SGA?

---

### Official Review · Reviewer_QYSj · 2024-10-31

**Soundness:** 2
**Presentation:** 2
**Contribution:** 2
**Rating:** 3
**Confidence:** 3

**Summary:**

The fundamental problem in causal inference is the inaccessibility of counterfactual outcomes, making it impossible to approach as a conventional supervised learning problem. Many approaches in the literature address the issue of estimating counterfactual outcomes by deriving and minimizing their upper bounds, thereby solving it indirectly. This paper proposes a method to find a tighter upper bound, resulting in an effective counterfactual prediction model. This is implemented through sub-treatment group alignment (SGA) and random temporal mask (RTM). SGA can be applied individually at each time step, while RTM applies to time-series data. The two methodologies were applied to tumor growth PK/PD data and MIMIC-III data, demonstrating superior performance compared to existing methods.

**Strengths:**

The approach presented in the paper is not an individual model but a universal methodology that can be applied to various existing models. As shown in Tables 1, 2, and 3, applying it to state-of-the-art models like CRN and CT resulted in better outcomes, demonstrating the broad applicability of the proposed methodology.

**Weaknesses:**

1. When looking closely at the two main methods of the paper, SGA and RTM, several questions arise. The questions raised in the section below need to be clearly addressed.

2. To read the main text of the paper, it is necessary to refer to the contents of the Appendix. This effectively exceeds the 9-page limit, utilizing more than 10 pages. The main paper should be complete, so all the necessary contents should be included in the main text of 9 pages.

**Questions:**

1. To perform SGA, GMM is used to attempt matching between each mixture component. Figures 1 and 2 show how the mixture components are matched and aligned in a one-dimensional schematic, making it easy to intuitively understand how they can be uniquely aligned in the order they are listed. However, since the representation transform $\phi$ function can be a one-to-one function that preserves multi-dimensionality, it is difficult to understand how more complex correspondences between different GMMs can occur and be aligned in dimensions higher than two. Looking at the equations in Lines 377 and 1035, we can see that the uniform mixture of distributions is performed in the calculations. How can the results be aligned as expected? Since GMM can yield different results depending on random initial values, does the uniform mixture calculation method ensure that SGA produces the same optimized results every time?

2. In Theorem 4.2, a small constant $\delta_c$ still exists in A2. Therefore, it seems difficult to say that the weighted sum on the left is always smaller than the W1 distance between the entire distributions on the right. What additional conditions are needed to say that the left provides a tighter upper bound? (I haven't checked all the proofs in the Appendix) Intuitively, the left term seems to be the sum of intra-cluster distances, while the right term includes both intra-cluster and inter-cluster distances. This gives me an intuitive understanding that the left should always be smaller than the right, and it is interesting that the left term acts as a new tighter upper bound in line 288. However, I do not understand why the constant term $\delta_c$ exists. The statement that the left term can act as a new tighter upper bound suggests that we only need to minimize intra-cluster distances without considering inter-cluster distances, which seems to more effectively reach the minimized point of counterfactual outcome estimation error. Please provide an intuitive explanation of Theorem 4.2 without reviewing the proofs in the Appendix.

3. RTM seems to be based on the assumption of some degree of continuity in time-series data. For example, tumor growth data. Even if data is missing at some time step, the missing data value can be roughly estimated. Would the RTM method be effective for data with insufficient continuity? Time-series data is usually assumed to be measured more densely and to have sufficiently long sequences, but would RTM be effective for short clinical trial data of about 12 weeks (measured biweekly, 6 times)? If not, it would be difficult to present RTM as a universal method, and a specific argument about the characteristics of the data where RTM is effective would be necessary.

4. What synergy effects do SGA and RTM have? They are applied separately in Tables 1 and 2, but how much better are the results when applied together? If they produce better results, it might be better to merge Tables 1 and 2 with the improved results into one table. Similarly, are there experimental results for SGA alone or RTM alone in Table 3? It might be better to display them together. If no synergy effects are observed, there seems to be no need to specify "time-series data" in the title. Only SGA can be applied to non-time-series data.

5. Clinical data includes a significant portion of categorical/binary data, which may cause the covariance matrices of mixture components to collapse when applying GMM. How can it be applied in such cases? Models like a mixture of factor analyzers (Richardson, Eitan, and Yair Weiss, On gans and gmms, NeurIPS 2018) might be able to replace GMM.

---

### Official Review · Reviewer_dixp · 2024-11-04

**Soundness:** 2
**Presentation:** 3
**Contribution:** 2
**Rating:** 5
**Confidence:** 4

**Summary:**

This paper proposes a method for counterfactual estimation in time series by using Sub-treatment Group Alignment (SGA) and Random Temporal Masking (RTM).

**Strengths:**

1. The studied problem is important and interesting, i.e., estimate counterfactual prediction under time series data.
2. The core idea is clearly explained and easy to follow. Including code improves reproducibility.
3. The paper is well written and theoretically sounds, proofs for all statements are provided in the appendix.

**Weaknesses:**

1. I would like to request some clarification on Figure 1. Is there a specific frequency for implementing SGA and RTM? Why is RTM only applied at T=2 and T=5?
2. I would like to seek clarification on the distinction between k and a. What is the relationship between the treatment group (a=0, a=1) and the sub-treatment group index k? Is k simply an index within each treatment group? Additionally, is there any restriction on the treatment type? I believe the proposed framework is designed for categorical treatments, but how would it perform with continuous treatments?
3. Is the use of Gaussian Mixture Models (GMMs) a restrictive assumption? I believe the framework relies on this approach. Please include some discussion on this point.
4. I am interested in understanding how effective your framework is. Specifically, as the level of confounding increases, does the difference in performance between the original framework and the one with added SGA and RTM also increase? (I cannot see this in Figure3(b)) Additionally, how does your framework perform with high-dimensional confounding variables?

**Questions:**

See Above.

---

### Meta-Review · Area_Chair_SV2t · 2024-12-19

**Metareview:**

This paper uses two techniques, Sub-treatment Group Alignment (SGA) and Random Temporal Masking (RTM), to improve counterfactual estimation in time-series data. The reviewers appreciated the clear problem formulation, theoretical foundation, and potential applicability of the methods across various models. However, they raised concerns about the robustness of experimental results, as findings were based on single runs without evaluating variability across multiple trials. The lack of real-world dataset validation, incomplete exploration of the synergy between SGA and RTM, and reliance on synthetic data further limit the practical impact. Additionally, key methodological justifications, such as the use of Gaussian Mixture Models, were insufficiently explained, and some theoretical aspects, like RTM’s foundation, were missing. While the authors’ rebuttal addressed some concerns, issues around robustness, real-world validation, and comprehensive evaluation remained unresolved.

**Additional Comments On Reviewer Discussion:**

There has been limited discussion for this submission, with one reviewer acknowledging the rebuttal. However, several key concerns raised by reviewers remain insufficiently addressed, despite the authors’ commitment to addressing them in the future: (1) the robustness of experimental results, as findings were based on single runs without evaluating variability across multiple trials; (2) the lack of validation on real-world datasets and the incomplete exploration of the synergy between SGA and RTM; and (3) insufficient methodological justifications, such as the choice of Gaussian Mixture Models, along with inadequate explanations of some theoretical aspects, including the foundation of RTM.

---

### Decision · Program_Chairs · 2025-01-22

Reject